# The International Virus Bioinformatics Meeting 2023

**DOI:** 10.3390/v15102031

**Published:** 2023-09-30

**Authors:** Franziska Hufsky, Ana B. Abecasis, Artem Babaian, Sebastian Beck, Liam Brierley, Simon Dellicour, Christian Eggeling, Santiago F. Elena, Udo Gieraths, Anh D. Ha, Will Harvey, Terry C. Jones, Kevin Lamkiewicz, Gabriel L. Lovate, Dominik Lücking, Martin Machyna, Luca Nishimura, Maximilian K. Nocke, Bernard Y. Renard, Shoichi Sakaguchi, Lygeri Sakellaridi, Jannes Spangenberg, Maria Tarradas-Alemany, Sandra Triebel, Yulia Vakulenko, Rajitha Yasas Wijesekara, Fernando González-Candelas, Sarah Krautwurst, Alba Pérez-Cataluña, Walter Randazzo, Gloria Sánchez, Manja Marz

**Affiliations:** 1European Virus Bioinformatics Center, 07743 Jena, Germanyana.abecasis@ihmt.unl.pt (A.B.A.); liam.brierley@liverpool.ac.uk (L.B.); simon.dellicour@ulb.be (S.D.); christian.eggeling@uni-jena.de (C.E.); santiago.elena@csic.es (S.F.E.); terence.jones@charite.de (T.C.J.); kevin.lamkiewicz@uni-jena.de (K.L.); gabriel.lencioni.lovate@uni-jena.de (G.L.L.); maximilian.nocke@ruhr-uni-bochum.de (M.K.N.); bernhard.renard@hpi.de (B.Y.R.); fernando.gonzalez@uv.es (F.G.-C.); alba.perez@iata.csic.es (A.P.-C.); wrandazzo@iata.csic.es (W.R.); gloriasanchez@iata.csic.es (G.S.); 2RNA Bioinformatics and High-Throughput Analysis, Friedrich Schiller University Jena, 07743 Jena, Germany; sarah.krautwurst@uni-jena.de; 3Global Health and Tropical Medicine, GHTM, Associate Laboratory in Translation and Innovation towards Global Health, LA-REAL, Instituto de Higiene e Medicina Tropical, IHMT, Universidade NOVA de Lisboa, UNL, Rua da Junqueira 100, 1349-008 Lisboa, Portugal; 4Department of Molecular Genetics, University of Toronto, Toronto, ON M5S 3E1, Canada; 5Donnelly Centre, University of Toronto, Toronto, ON M5S 1A1, Canada; 6Leibniz Institute of Virology, Department Viral Zoonoses—One Health, 20251 Hamburg, Germany; sebastian.beck@leibniz-liv.de; 7Department of Health Data Science, University of Liverpool, Liverpool L69 3GF, UK; 8Spatial Epidemiology Lab (SpELL), Université Libre de Bruxelles, CP160/12, 50 av. FD Roosevelt, 1050 Bruxelles, Belgium; 9Laboratory for Clinical and Epidemiological Virology, Department of Microbiology, Immunology and Transplantation, Rega Institute, KU Leuven, University of Leuven, 3000 Leuven, Belgium; 10Institute of Applied Optics and Biophysics, Friedrich Schiller University Jena, Max-Wien-Platz 1, 07743 Jena, Germany; 11Institute for Integrative Systems Biology (I2SysBio), CSIC-Universitat de Valencia, Catedratico Agustin Escardino 9, 46980 Valencia, Spain; 12Institute of Virology, Charité, Universitätsmedizin Berlin, Charitéplatz 1, 10117 Berlin, Germany; 13Department of Biological Sciences, Virginia Tech, Blacksburg, VA 24061, USA; 14The Roslin Institute, University of Edinburgh, Edinburgh EH25 9RG, UK; 15Department of Zoology, University of Cambridge, Downing Street, Cambridge CB2 3EJ, UK; 16Max-Planck Institute for Marine Microbiology, Celsiusstraße 1, 28359 Bremen, Germany; 17Paul-Ehrlich-Institut, Host-Pathogen-Interactions, 63225 Langen, Germany; 18Department of Genetics, School of Life Science, The Graduate University for Advanced Studies (SOKENDAI), Mishima 411-8540, Japan; 19Human Genetics Laboratory, National Institute of Genetics, Mishima 411-8540, Japan; 20Department for Molecular & Medical Virology, Ruhr University Bochum, 44801 Bochum, Germany; 21Digital Engineering Faculty, Hasso Plattner Institute, University of Potsdam, 14482 Potsdam, Germany; 22Department of Microbiology and Infection Control, Faculty of Medicine, Osaka Medical and Pharmaceutical University, Osaka 569-8686, Japan; shoichi.sakaguchi@ompu.ac.jp; 23Institute for Virology and Immunobiology, University of Würzburg, Versbacher Str. 7, 97078 Würzburg, Germany; 24Computational Genomics Lab., Department of Genetics, Microbiology and Statistics, Institut de Biomedicina UB (IBUB), Universitat de Barcelona (UB), 08028 Barcelona, Spain; 25Martsinovsky Institute of Medical Parasitology, Tropical and Vector Borne Diseases, Sechenov First Moscow State Medical University, 119991 Moscow, Russia; 26Institute for Bioinformatics, University of Medicine Greifswald, Felix-Hausdorff-Str. 8, 17475 Greifswald, Germany; 27Joint Research Unit “Infection and Public Health” FISABIO, University of Valencia, 46010 Valencia, Spain; 28VISAFELab, Department of Preservation and Food Safety Technologies, Institute of Agrochemistry and Food Technology, IATA-CSIC, 46980 Valencia, Spain; 29German Center for Integrative Biodiversity Research (iDiv) Halle-Jena-Leipzig, 04103 Leipzig, Germany; 30Michael Stifel Center Jena, Ernst-Abbe-Platz 2, 07743 Jena, Germany; 31Cluster of Excellence Balance of the Microverse, Friedrich Schiller University Jena, 07745 Jena, Germany; 32Leibniz Institute for Age Research—Fritz Lippman Institute, 07745 Jena, Germany

**Keywords:** bioinformatics, tools, machine learning, bacteriophages, virus discovery, virus classification, virus visualization, viral infection, viromics, molecular epidemiology, phylodynamic analysis, RNA viruses, viral sequence analysis, viral surveillance, metagenomics

## Abstract

The 2023 International Virus Bioinformatics Meeting was held in Valencia, Spain, from 24–26 May 2023, attracting approximately 180 participants worldwide. The primary objective of the conference was to establish a dynamic scientific environment conducive to discussion, collaboration, and the generation of novel research ideas. As the first in-person event following the SARS-CoV-2 pandemic, the meeting facilitated highly interactive exchanges among attendees. It served as a pivotal gathering for gaining insights into the current status of virus bioinformatics research and engaging with leading researchers and emerging scientists. The event comprised eight invited talks, 19 contributed talks, and 74 poster presentations across eleven sessions spanning three days. Topics covered included machine learning, bacteriophages, virus discovery, virus classification, virus visualization, viral infection, viromics, molecular epidemiology, phylodynamic analysis, RNA viruses, viral sequence analysis, viral surveillance, and metagenomics. This report provides rewritten abstracts of the presentations, a summary of the key research findings, and highlights shared during the meeting.

## 1. Introduction

The International Virus Bioinformatics Meeting (ViBioM) was the sixth edition of the virus bioinformatics meeting organized by the European Virus Bioinformatics Center (EVBC). The EVBC was founded in 2017 to bring together experts in virology and virus bioinformatics in Europe [1,2]. The EVBC is constantly growing, having currently 276 members (~12% increase since the last meeting in 2022 [3]) from 158 research institutes distributed over 41 countries worldwide. ViBioM 2023 took place in Valencia, Spain from 24–26 May 2023. From all registered participants, ~23% are EVBC members; thus, ViBioM is attracting scientists far beyond the EVBC community. In contrary to 2022, the participants had a highly interactive scientific environment by face-to-face interactions. In total, the meeting featured 8 invited and 19 selected talks in eleven sessions on three days, as well as 74 posters, which were presented during three poster sessions.

## 2. Scientific Program

A number of high-quality presentations were given by leading experts and junior scientists on several different topics in virus bioinformatics. From 61 submissions (a ~17% increase compared to 2022 [3]), we selected 21 talks (acceptance rate: ~34%; two of them have been merged, and one has been invited for a keynote). Due to the high amount of submissions on SARS-CoV-2-related research, we decided to add an additional conference day. On the first day, we were focusing on five topics in five sessions: Phages (see Section 2.1), Virus discovery and classification (see Section 2.2), Virus visualization (see Section 2.3), Viral infection (see Section 2.4), and Viromics (see Section 2.5). On day two, we aimed for three sessions, namely: Molecular epidemiology and phylodynamic analyses (see Section 2.6), RNA viruses: structure and evolution (see Section 2.7), and  Viral sequence analysis (see Section 2.8). Finally on the third day, we explored the fields of: Machine learning in viral surveillance (see Section 2.9), Viral pathogenesis (see Section 2.10), and  Metagenomics for Identifying and Tracking Potential Zoonotic Viruses (see Section 2.11). Lara Fuhrmann (VILOCA: Sequencing quality-aware haplotype reconstruction and mutation calling for short- and long-read data) was selected for Best ECR Talk Award. During three poster sessions, 74 posters were presented. Two presenters were selected for the Best Poster Award: Muriel Ritsch (Non-retroviral RNA viruses integrated into the human genome) and Jordi Sevilla (Tracking intra host evolution of SARS-CoV-2).

### 2.1. Phages

Phages are relevant to all human beings due to their abundance, predation, disease control potential, influence on the gut microbiome, biotechnological applications, and contributions to genetic diversity and evolution. Their study and utilization hold great promise for addressing challenges in healthcare, agriculture, and environmental sustainability. Beside an awe-inspiring and fantastic presentation by Robert Edwards, we recieved also a brilliant presentation by Yasas Wijesekara, who is also the travel award winner for this year. This session was chaired by EVBC member Anca Segall (San Diego State University, San Diego, CA, USA).

#### Jaeger: A Deep Learning Approach for Predicting Bacteriophage Sequences in Metagenomic Data (by Rajitha Yasas Wijesekara)

Microbial communities are complex admixtures containing vastly different organisms representing the three domains of life and their viruses. Bacteriophages, the viruses that infect bacteria, are ubiquitous in almost every environment and play a crucial role in shaping the ecological and evolutionary processes of ecosystems by controlling bacterial abundances [4,5,6,7]. They also influence bacterial phenotype in the virocell state by altering bacterial metabolism and drive global nutrient flow [8,9]. Detecting phages in metagenomic datasets requires specialized bioinformatic tools. Wijesekara et al. presented Jaeger https://github.com/Yasas1994/Jaeger, a novel artificial intelligence (AI) tool for predicting bacteriophage sequences in metagenomic data that can be applied to individual reads (70% accuracy), assembled contigs (90% accuracy), and bins (93% accuracy). Additionally, Jaeger can detect prophages (74% accuracy), and identify other sequence categories such as eukaryotic, bacterial, and archaeal genomes. Jaeger utilizes a deep learning model with dilated convolution and residual connections that learn feature representations from nucleotide sequences, which are subsequently used for classification. The authors demonstrate that their novel neural architecture performs better than other available methods. Specifically, they compared the performance of Jaeger to PPRMeta, DeepVirFinder, Seeker, and VirSorter2 on the phages in the IMGVR (v4) database and real metagenomic datasets from three different biomes, showing 10–35% decrease in false positive rate without compromising on sensitivity [10,11,12,13,14,15]. Together, Jaeger adds a new AI-powered tool to the metagenomics toolbox that will help to understand the composition of complex communities from metagenomic sequencing.

### 2.2. Virus Discovery and Classification

Virus discovery and classification are fundamental aspects of virology. Through ongoing research and technological advancements, new viruses are continuously being identified. Classification involves categorizing viruses based on their genetic material, structure, and mode of replication. This knowledge aids in understanding viral evolution, transmission, and the development of diagnostic tools and antiviral strategies. This session was chaired by Justine Charon (University of Syndey, Australia).

#### 2.2.1. Illuminating the RNA Virome through Ultra-Massive Sequence Analysis (by Artem Babaian)

Transcriptomic/metatranscriptomic sequencing is revolutionizing the exploration of Earth’s virome. Yet analysis methods are inefficient and don’t scale to the available data. The global biology community has freely shared >30 petabases (3 $\cdot 10^{16}$  nt) of sequence data from 10+ million biological samples [16,17]. Painstakingly collected over 15 years, public data encompass all continents, oceans, thousands of animals, plant, and fungal species, and estimated [16,18] to be valued at $3.6–14.9 billion dollars in direct sequencing cost.

To uncover the total diversity of RNA viruses, Babaian et al. developed a cloud-based sequence alignment platform called Serratus www.serratus.io, with which they analyzed 7.4 million public sequencing datasets for the RNA viral hallmark gene, RNA-dependent RNA polymerase [19]. They identified the equivalent of >300,000 novel RNA viruses which is over an order of magnitude increase of known virus diversity; including at least nine new species of Coronaviruses (CoV).

Planetary-scale informatics provide unprecedented depth of insights with which to describe virus evolution and ecology. Seven of the novel CoV identified are encoded on segmented genomes (of monophyletic origin). The majority of novel nidoviruses were identified in samples from aquatic vertebrates (axolotl, leopard frog, seahorse, fugu fish, etc.) supporting that there is a vast uncharacterized reservoir of marine nidoviruses. With a high-sensitivity search, Babaian et al. detect genome-fragments from >40 distinct uncharacterized nidoviruses. For both novel and known nidoviruses, the authors expand virus host-ranges, geographic distributions, ecological niches and identify potential hidden reservoirs in an unbiased manner which at times, challenge the preconceptions about nidovirales [19].

Explosive growth in data volumes and the breadth of Earth’s biodiversity captured by global biological sequencing, see Figure 1, will change bioinformatics in the near future. The next decade of virology will be illuminated by computational advances and requires a shift in our conceptualization of virus discovery and it’s applications to pandemic surveillance. What can be learned today (with scalable methods) from Nidovirales lays the foundation for how we will explore the 100+ million virus species we project to discover by 2030.

#### 2.2.2. RNA Virus Discovery Using HMM of Large-Scale RNA-Dependent RNA Polymerase Sequence Data: NeoRdRp 2.0 (by Shoichi Sakaguchi)

RNA-dependent RNA polymerase (RdRp) is unique to RNA viruses and serves as a critical marker in searching RNA viruses from RNA sequencing data. Since virus detection methods rely on sequence similarity, it is difficult to identify viruses whose similar sequences are unavailable in the database. Therefore, Sakaguchi et al. developed an analysis pipeline that leverages a hidden Markov model (HMM) to enhance the detection of RdRp sequences with low similarity. The authors utilized three reported RdRp amino acid sequence datasets (hereafter called the ‘seed datasets’): the 4620 sequences by Wolf et al. [20], the 14,680 sequences by Edgar et al. [19], and the 209,588 sequences by Zayed et al. [21]. Additionally, Sakaguchi et al. used 18,790 sequences from RNA viruses registered in the NCBI Virus database as a ‘bulk virus dataset’ and 565,928 sequences registered in UniProtKB as a ‘test dataset’. The analysis pipeline began with clustering based on sequence similarity within the seed datasets, followed by a multiple sequence alignment for each cluster (see Figure 2). The authors defined a domain as a region where at least 75% of the entire sequence was aligned and created an HMM profile for each domain. They then applied the HMM profiles to the hmmsearch and detected RdRp sequences from the bulk virus dataset. Finally, they constructed the HMM profile NeoRdRp https://github.com/shoichisakaguchi/NeoRdRp from the RdRp domain dataset obtained from seed datasets and the RdRp domain sequences derived from the hmmsearch of the bulk virus dataset. The NeoRdRp yielded 13,038 HMM profiles, which encompassed known RdRp motifs. Sakaguchi et al. applied these HMM profiles to perform a hmmsearch of the test dataset and successfully detected 832 out of 836 RdRp sequences. The accuracy and specificity were 99.5% and 79.9%, respectively. It is planned to continually refine the system by updating and searching RNA sequencing data, aiming to utilize the system to detect novel RNA viruses. Version 1.1 has been reported in a paper [22].

#### 2.2.3. Automated Classification of Giant Virus Genomes Using Protein Family Barcodes (by Anh Ha)

Large DNA viruses of the phylum Nucleocytoviricota, or ‘giant viruses’, are ubiquitous in the environments and play important roles in shaping the dynamics of global ecosystems. Documented members of the phylum are highly diverse and can be partitioned into six orders and up to 32 potential families [23]. Giant viruses are known to infect a broad range of eukaryotic hosts and appear to have undergone multiple gene exchanges with their hosts [24]. As a result, their genomes often carry diverse genes involved in various cellular processes, such as the TCA cycle, translation, light sensing, and cytoskeletal dynamics [25,26,27]. Due to the large phylogenetic breadth of this viral group and the highly complex, chimeric nature of their genomes, taxonomic classification of giant viruses, particularly incomplete metagenome-assembled genomes (MAGs) could be challenging. Here Ha et al. utilized a machine learning approach to predict the taxonomic classification of novel giant virus MAGs based on profiles of protein family content, which they refer to as protein family barcodes. They applied random forests to a training set of 1503 quality-checked, phylogenetically diverse Nucleocytoviricota genomes using a pre-selected set of giant virus orthologous groups (GVOGs). The classification model was predictive of giant viruses’ taxonomic group with a cross-validation accuracy of 99.5% to the Order level and 97% to the Family level. The authors observed that no individual GVOGs or genome features were critical to the algorithm’s performance and the model’s predictions, suggesting that classification predictions were based on a broad genomic signature, which lessened the necessity of a fixed set of marker genes for taxonomic assigning purposes. Their classification model was validated with an independent test set of 814 giant virus genomes with varied genomic completeness and predicted taxonomy with 97.5% and 95% accuracy to the Order and Family level, respectively. The results provide a fast and accurate method for the classification of giant viruses that could easily be adapted to other viral groups.

#### 2.2.4. Using gb2seq to Work with Unannotated Viral Genomes Based on a GenBank Reference (by Terry Jones)

The talk by Jones et al. presented gb2seq https://github.com/VirologyCharite/gb2seq, a Python library and associated command-line scripts that derive information regarding unannotated viral genomes (e.g., a consensus called from a BAM file following alignment with bwa or bowtie), based on annotations in a GenBank reference. The library provides for: extraction of aligned features as nucleotide or amino acid sequences; retrieving information about what is at a site (features, nucleotide, amino acid, codon, frame, etc); translation of offsets between the reference and the unannotated genome with offsets that are absolute or relative to a feature; detailed alignment information for features; use of different aligners (MAFFT and edlib, currently); JSON annotations of the features of an unannotated genome; and convenience methods for checking genomes for sets of expected nucleotide or amino acid values. The command-line scripts provide a simple interface to the library functions, e.g., to extract a translated feature from a set of genomes and check for how many expected substitutions are present.

### 2.3. Virus Visualization

The integration of genomic information and imaging techniques offers valuable insights into viral biology. By combining approaches such as Fluorescence In Situ Hybridization (FISH), Genome-Wide Association Studies (GWAS), and multi-omics integration, researchers can gain a deeper understanding of viruses at a spatial and molecular level. Bioinformatics tools assist in automated analysis of imaging data, aiding in the identification and quantification of viral structures. This integration enhances our understanding of viral biology, mechanisms of viral pathogenesis, and facilitates the development of targeted interventions and therapeutics. Manja Marz (FSU, Jena, Germany) has invited Christian Eggeling to give a presentation on his research area with the aim to open a pathway for research combining virus omics data and morphological features.

#### Advanced Optical Microscopy of Virus-Cell Interactions: Challenges and Potentials (by Christian Eggeling)

Understanding virus infectivity also requires revealing molecular interaction details during the virus-host interaction. For example, an increased mobility and subsequent aggregation of envelope proteins during maturation of HIV-1 virions drives their potential to dock host-cell receptors and thus infectivity, as revealed by optical microscopy [28,29]. Unfortunately, the direct and non-invasive observation of the interactions in the living cell membrane is often impeded by principle limitations of conventional far-field optical microscopes, for example with respect to limited spatio-temporal resolution. However, the advent of super-resolution microscopy techniques has created unique opportunities of investigating the organization and mobility of viral and cellular molecules at the required small spatial scales [30,31]. Taking HIV-1 and SARS-CoV-2 as examples, Eggeling has presented how such advanced optical microscopy approaches help highlighting novel aspects of virus-membrane interactions (Figure 3) and what challenges one still faces. An important issue will be, how these microscopy-based information can be correlated with bioinformatic data to boost information content. Presenting the potential of microscopy to bioinformatics at the ViBiom 2023 has done a first step.

### 2.4. Viral Infection

Viral infections are a significant public health concern, causing a wide range of diseases in humans and animals. Understanding viral infections is crucial for developing effective prevention strategies, diagnostics, and therapeutics. Bioinformatics plays a vital role in the analysis of viral infections by integrating large-scale genomic, transcriptomic, and proteomic data to identify viral genomes, study their evolution, and predict their pathogenic potential. The session has been chaired by Anamarija Butkovic (Institut Pasteur, Paris, France).

#### 2.4.1. SARS-CoV-2-Host Interactions at the Single-Cell Level: A Dynamical Complex Systems Approach (by Santiago F. Elena)

Elena et al. presented an analysis of the response of three different cell types during the progression of SARS-CoV-2 infection [32]. To do so, they have performed a meta-analysis of three publicly available single-cell RNA sequencing (scRNA-seq) datasets obtained from in vitro inoculation studies of human bronchial epithelial cells [33], and colon and ileum organoids [34]. ScRNA-seq has become a powerful technique to study the dynamic changes in the transcriptome of a population of cell subjected to an external stimulus. Indeed, different cells in the population are found at different stages of infection, as shown by the differences in viral reads that they contain. Therefore, a bulk analysis is clearly inappropriate to identify dynamical gene responses associated to infection. Instead, the authors have classified cells according to their infection status using virus accumulation as a proxy to time, see Figure 4. This approach has revealed that about 90% of genes exhibited a transcriptional response characterized by a triphasic pattern comprising an early down-regulatory phase at the beginning of infection, followed by a transient massive up-regulation and, finally, a final down-regulation as cells’ viability declines, see Figure 4. Focusing in the 10% of genes that significantly deviate from this canonical triphasic behavior, Elena et al. found an enrichment in genes related to immune responses, translation and mitochondrial oxidative activities. Interestingly, their analyses have shown that the transcriptional profiles of genes encoding for the main intracellular sensors that recognize double-stranded RNAs and trigger the innate immune response upon infection, MDA-5 and RIG-I, do not deviate from the majoritarian response. This observation suggests that the transcriptional shutdown of the IFN response induced by MDA-5 and RIG-I is probably not specific, and results from a more general mechanism affecting most genes in the infected cell. Additionally, a correlation network analysis revealed a distinct correlated response of mitochondrially-encoded genes and genes involved in translation. Finally, the authors proposed a mechanistic model to explain this dynamic profiles. The key player in the model is viral protein Nsp1, that has been shown to block the export of specific mRNAs from the nucleus [35,36]. Expression of nsp1 acts like an off-switch for Nsp1-sensitive transcripts (e.g., interferon genes) that results in their accumulation in the nucleus while Nsp1-insensitive transcripts can transit normally to the cytoplasm. The middle phase corresponds to the inactivation of the Nsp1-induced blockage, which releases the stalled transcripts to the cytoplasm that results in the increased transcription levels observed during the middle phase. Finally, the late phase corresponds to a shutdown of transcription that also results in a global decrease in transcripts levels.

#### 2.4.2. Metabolic Labeling, Time Series, and Single Cells: A Multifaceted Approach to Studying Infection
(by Lygeri Sakellaridi)

Viral infection is a dynamic biological process, facilitated by complex interactions between host and viral genes. On a cellular level, infection may result in different outcomes, e.g., lytic or latent. Elucidating the molecular mechanisms that determine the infection outcome is a challenging problem. In order to tackle this problem, Sakellaridi et al. proposed an approach that combines single cell RNA sequencing with metabolic RNA labeling. Metabolic labeling utilizes 4-thiol-uridine (4sU) which is introduced into cells and subsequently incorporated into newly transcribed RNA. Before sequencing, RNA is subjected to chemical treatment that results in the conversion of 4sU into cytosine or cytosine analogs [37,38,39]. This allows distinguishing between RNA that was synthesized before and after the start of labeling based on the presence of T-C mismatches in the newly transcribed reads. Accurate quantification of new and pre-existing reads is hindered by the low incorporation rate of 4sU in cells [37], as well as the presence of naturally occuring mismatches in reads. To address these challenges, the authors’ lab developed GRAND-SLAM https://github.com/erhard-lab [40]: a statistical approach that uses a binomial distribution to provide unbiased estimates of the percentage of labeled RNA per gene, as well as its posterior distribution that represents uncertainty of quantification. This output can be incorporated in downstream analyses, such as estimation of RNA kinetics and changes in gene expression. To that end, Sakellaridi et al. developed grandR [41], a computational package that facilitates such analyses while taking advantage of the uncertainty estimates. Metabolic labeling of single cells makes it possible to acquire two snapshots of expression per cell in a population, corresponding to the transcriptomic state before (past state) and after (current state) the end of labeling. Heterogeneity in the past state can be used to make potentially causal inferences in the current state: e.g., anti-correlation of expression of a given gene in the past state against the total viral gene expression in the current state would indicate an anti-viral gene. Extending this reasoning, time courses allow following the fate of cells over extended periods of time. Sakellaridi et al. introduce a novel trajectory inference approach for connecting cells from consecutive time points based on similarity between their past and current transcriptomic profiles. The authors envision the data as a network where cells are connected across time, and their changing expression trajectories ’flow’ through the network. Mathematically, this is equivalent to the minimum cost-maximum flow problem and can be solved by a well-established optimization algorithm [42]. Solving the network results in expression trajectories that reflect the different paths a biological process may follow across time. By constructing such trajectories in a population of infected cells, it becomes possible to determine factors that influence the infection outcome.

### 2.5. Viromics

Viromics heavily relies on computational tools for the comprehensive analysis of viral genomes and metagenomic data. These tools encompass a wide range of bioinformatics algorithms and pipelines that aid in various stages of viromic analysis, including sequence quality control, read assembly, viral genome binning, taxonomic classification, and functional prediction. Computational approaches are essential for unraveling the complexity of viromic datasets, extracting valuable insights about viral diversity, evolution, and interactions within microbial communities. These tools enable the identification of novel viral species, characterization of viral genes and functional elements, exploration of viral-host interactions, and the potential discovery of new antiviral targets. Continuous advancements in computational tools and techniques are crucial for enhancing our understanding of the virosphere and its implications in diverse fields, from environmental microbiology to human health. Our viromics session has been hosted by Spyros Lytras (MRC, Glasgow, UK).

#### 2.5.1. Ancient Virome Analyses Using Metagenomic Data from Ancient Individuals
(by Luca Nishimura)

Ancient DNAs have been discovered from various kinds of archeological samples such as bones and mummified tissues. Those ancient samples contain ancient viral genomes which existed in ancient organisms’ bodies [43]. Those ancient viral genomes are useful to elucidate past pandemic events and long-term viral evolution. For instance, human pathogenic viruses such as influenza A virus and hepatitis B virus have been analyzed since 1997. However, the number of ancient viruses thus far identified is small; most are human pathogenic viruses. Nishimura et al. analyzed ancient people’s whole genomic sequencing (WGS) data to discover ancient viruses more comprehensively. They utilized genomic data from 36 ancient individuals who dwelled in the Japanese archipelago and also more than 300 publicly available data. Firstly, the authors conducted *de novo* assembly of non-human reads to obtain longer contigs. Those contigs were used for homology search against the modern viral reference genomes and other methods such as machine-learning methods and bacterial immunological memories to find viral genomes. As a result, more than 50,000 candidates of ancient viral contigs were detected, including about 200 high-quality ancient viral genomes. To characterize ancient virome, Nishimura et al. analyzed ORF components and phylogenetic relationships based on those contigs. For example, they obtained the nearly complete sequence of the Siphovirus contig89 (CT89) existing in the human oral environment [44]. The authors estimated the relationships between the modern CT89 and the most recent common ancestor by phylogenetic analyses. They then compared the viral components in each sample showing the differences between ancient and modern samples which might reflect different dietary characteristics. Those results suggest that the metagenomic data from ancient samples are useful for elucidating ancient viral characteristics and long-term viral evolution.

#### 2.5.2. One’s Trash Is Another’s Treasure—Mining Viromics Datasets for Traces of EV Mediated Horizontal Gene Transfer (by Dominik Lücking)

Marine environmental viral metagenomes, commonly referred to as ‘viromes’, are typically generated by physically separating viral-like particles (VLPs) from the microbial fraction based on their size and mass. However, most of the methods used to enrich extracellular vesicles (EVs) and gene transfer agents (GTAs) simultaneously [45,46]. Consequently, the sequence space traditionally referred to as a ‘virome’ contains host-associated sequences, transported via EVs or GTAs. Lücking et al. therefore propose to call the genetic material, isolated from size-fractionated (0.22 μm) and DNase treated samples, protected environmental DNA (peDNA), Figure 5. This sequence space contains viral genomes, DNA transduced by proviruses and DNA transported in EVs and GTAs. Since there is currently no definitive genetic signature for EV-transported DNA, scientists rely on the successful removal of contaminating remaining cellular and free DNA, when analyzing peDNA. Using marine samples collected from the North Sea, the authors generated a thoroughly purified peDNA dataset and developed a bioinformatic pipeline to determine the potential origin of the purified DNA. This pipeline (https://github.com/dluecking/mviest) was applied to their dataset as well as existing global marine ‘viromes’, enabling the identification of known GTA and EV producers, and organisms with actively transducing proviruses as the source of the peDNA, thus confirming the reliability of their approach. Additionally, Lücking et al. identified novel and widespread EV producers and found quantitative evidence suggesting that EV-mediated gene transfer plays a significant role in driving horizontal gene transfer (HGT) in the world’s oceans.

### 2.6. Molecular Epidemiology and Phylodynamic Analyses

Molecular epidemiology and phylodynamic analysis are two interconnected fields that utilize computational tools to study the spread and evolution of infectious diseases. Molecular epidemiology combines molecular biology and epidemiology to investigate the transmission dynamics, source attribution, and genetic characteristics of pathogens. Phylodynamic analysis involves the use of computational models and phylogenetic methods to infer the evolutionary history, population dynamics, and transmission patterns of pathogens based on genetic data. Computational tools play a crucial role in processing large-scale genomic datasets, reconstructing phylogenetic trees, estimating evolutionary rates, detecting transmission clusters, and identifying key determinants of disease emergence and spread. These tools enable researchers to gain insights into the factors driving disease outbreaks, design effective control strategies, and contribute to global health surveillance efforts. This session was hosted by Francesca Young (MRC, Glasgow, UK).

#### 2.6.1. HIV-1 Transmission Studies Using Phylogenetics: Can Evolution Help Guide Public Health Decisions? (by Ana Abecasis)

Since the 1990s, HIV-1 transmission chains reconstruction has been used for different purposes. It’s first use for court cases of HIV-1 transmission has led to important conceptual and ethical discussions [47]. Later on, HIV-1 transmission chains reconstruction combined with socio-demographic and behavioral data has been used frequently in the context of public health epidemiological studies [48]. By combining socio-demographic, behavioral and clinical data, Abecasis et al. reconstructed transmission chains in different contexts to better understand the most important determinants of transmission of HIV-1 infection in each scenario. They addressed potential pitfalls and methodological constraints, and presented the main challenges, implications and applicability to future studies. The authors’ results indicate the importance of its use to complement classical epidemiological approaches, indicating routes of transmission of HIV-1 infection in migrants living in Portugal [49] and transmission routes more associated with transmission of antiretroviral drug resistance [50].

#### 2.6.2. Molecular Epidemiological Approaches to Investigate the Dispersal Dynamic of Viruses and the Environmental Factors Impacting It (by Simon Dellicour)

Recent advances in genomics, mathematical modelling and computational biology have enabled molecular approaches to become key methods to investigate the spread of viral infectious diseases. In the emerging field of molecular epidemiology, genetic analyses of pathogens are used to complement traditional epidemiological methods in various ways. For instance, genetic analyses offer the possibility to infer linkages between infections that are not evident without analyzing viral genomes. In particular, the development of phylogeographic methods has enabled to reconstruct dispersal history of epidemics in a discretised or on a continuous space, using only a relatively limited number of viral sequences sampled from known locations and times. At the Spatial Epidemiology Lab (SpELL, ULB) https://github.com/sdellicour, Dellicour et al. develop and apply new analytical approaches exploiting such phylogeographic reconstructions to test epidemiological hypotheses about the external and environmental factors impacting the dispersal history and dynamic of viral epidemics [51,52,53].

#### 2.6.3. Phylodynamic Analysis of A(H5N1) Highly Pathogenic Avian Influenza Viruses Provides Insight into Movement Dynamics and Host Specificity (by Will Harvey)

Phylodynamic analyses, modelling approaches based on co-analysis of genetic sequences and associated metadata within a phylogenetic framework, can shed insight on the characteristics and drivers of outbreaks of infectious diseases. Since 2021, highly pathogenic avian influenza (HPAI) viruses of subtype A(H5N1) have caused a panzootic of unprecedented scale. This has affected both wild and domestic birds with high mortality outbreaks in atypical host species such as shorebirds. Spill-overs in a diverse range of mammalian species including die-offs involving thousands of pinnipeds have raised concerns over zoonotic potential. These viruses can readily exchange genomic segments with local low pathogenic avian influenza (LPAI) viruses via reassortment, a mechanism that can facilitate dramatic phenotypic change. Observational data suggest changes in the seasonality of recent HPAI A(H5N1) viruses and relaxation of host specificity, however virological or ecological explanations remain elusive.

Using phylodynamic approaches, it is possible to reconstruct spatio-temporally resolved phylogenetic trees and examine trends relative to pre-panzootic A(H5N8) viruses. Examination of patterns in the timing and locations of reassortment events shows variation in the evolutionary trajectories of reassortant lineages. Exploration of possible changes in host preference or specificity using tree-based metrics to estimate fitness in different host types shows, for example, lineage adapted to spread in birds of the avian orders Charadriiformes, that acquired gene segments LPAI H13 viruses. This demonstrates phenotypic diversification among recent HPAI A(H5N1) viruses, see Figure 6. Phylogenetic discrete trait modelling indicates this modification in host specificity coincides with changes in the times and locations of virus movements such as those between Great Britain and continental Europe. Such diversification has consequences for spill-over opportunities at the wildlife-domestic poultry interface. Furthermore, a relaxation of host tropism increases the likelihood of further reassortment events increasing the potential for emergence of novel genotypes with changed characteristics.

### 2.7. RNA Viruses: Structure and Evolution

RNA secondary structures play a crucial role in the function and replication of RNA viruses, and computational tools are invaluable for their analysis. These tools employ algorithms and predictive models to identify and predict RNA secondary structures based on sequence information. They help in identifying conserved structural motifs, such as stem-loops and pseudoknots, that are important for viral replication, translation, and packaging. Computational tools also aid in studying RNA-RNA and RNA-protein interactions within the context of viral infections. By analyzing RNA secondary structures, researchers can gain insights into the mechanisms of viral pathogenesis, identify potential targets for antiviral interventions, and design novel therapeutics. These computational approaches greatly enhance our understanding of RNA virus biology and contribute to the development of effective strategies to combat viral diseases. The session was hosted by Dmitrij Frishman (TUM, Munich, Germany).

#### 2.7.1. Viral RNA Secondary Structures: Canonical and Beyond (by Kevin Lamkiewicz & Sandra Triebel)

In recent years, RNA biology has evolved around identifying and annotating functional (non-)coding RNAs in organisms from all domains of life. Especially for RNA viruses, it is conceivable that genomic regions and transcripts serve additional functions essential for viral replication induced by their RNA secondary structures. In the canonical RNA structure model, base-pairing of complementary nucleotides will fold the RNA into local and global structural elements, such as the prominent stem-loop hairpin structure.

Critical assessment of RNA secondary structures within viral genomes has led to several molecular insights for viruses. For example, the internal ribosome entry site (IRES) of the Hepatitis C virus (HCV) is an RNA structure within the 5${}^{\prime }$ UTR of the genome. It mediates the cap-independent translation initiation of viral proteins [54]. The 3${}^{\prime }$ UTR in flaviviruses exposes, among others, exoribonuclease resistant RNA structures (xrRNAs), prolonging viral genomes within the hosts’ cytoplasm [55,56]. Another well-described example is the UTRs of coronaviruses. Here, several stem-loops are seemingly crucial for viral transcription, translation, and replication [57,58,59]. Despite well-characterized RNA structures in untranslated regions of human-infecting RNA viruses, structures within coding sequences remain unclear. For example, it is hypothesized that RNA-RNA interactions play an essential role in the discontinuous transcription mechanism of coronaviruses. For Influenza A virus, one debated hypothesis is that the packaging of the eight segments is mediated via RNA-RNA interactions [60,61,62].

Therefore, the determination and functional analysis of structural RNA elements promises many novel insights into the viral life cycle and further new targets for therapeutic approaches and drugs. Bioinformatic models and tools are used to assess probable and kinetically favorable RNA foldings to facilitate and increase our understanding of RNA secondary structures in viral genomes. Multiple sequence alignments (MSAs) are commonly used to confirm the importance of functional regions via evolutionary conservation. Structural conservation indicated by compensatory mutations is considered with structure-guided MSAs proposed by Sankoff (implemented in, e.g., LocARNA [63]). To evaluate whether such tools directly apply to viral genomes and transcripts, Lamkiewicz and Triebel used LocARNA on a prominent region within the genome of *Filoviridae*. The transcription start site (TSS) is a conserved sequential motif upstream of each ORF in *Filoviridae*. Additionally, to sequence conservation, it is further embedded in an RNA structure which has been well-described in the literature [64,65].

The authors assessed the performance of an automated RNA secondary structure prediction method. The structure-guided MSA calculated by LocARNA is shown in Figure 7A. While structural conservation looks promising, the sequential motif is disrupted for the representative *Dianlovirus*. However, careful manual curation reveals that the TSS motif can be recovered, leading to less sequential diversity in alignment columns involved in base pairings, see Figure 7B. While software development is going in the right direction, the findings underscore the importance of critically evaluating the results obtained from automated tools and highlight the significance of expert intervention in refining and interpreting complex biological data.

#### 2.7.2. Recombination and Modular Evolution of Positive-Strand RNA Viruses: Similar, but Not the Same
(by Yulia Vakulenko)

Recombination is very common in positive-strand RNA viruses. Along with a high mutation rate, it is one of the major forces generating genetic diversity [66]. Vakulenko et al. systematically analyzed patterns of natural recombination in four (+)RNA virus families–*Astroviridae, Caliciviridae, Picornaviridae* and *Coronaviridae*, using both classical recombination detection methods [67,68] and by comparing correspondence of genetic distances in different genome regions [69]. The authors developed an R package, recDplot https://github.com/v-julia/recDplot, for visualizing recombination in viral sequences. A common (and generally known) feature of these virus families was frequent recombination between genome regions encoding nonstructural and structural proteins [67,70,71,72]. However, the recombination profiles within these genome regions were contrasting. In picornaviruses, there was frequent recombination within the nonstructural genome region with no prominent hotspots and almost absent recombination within the structural genome region. Caliciviruses routinely exchanged full structural and nonstructural blocks of the genome, but had few, if any, recombination events within these regions [73]. In astroviruses, moderate recombination was observed within both structural and nonstructural genomic regions. In coronaviruses, the spike gene, but not other structural proteins genes (E, M, N), was most commonly exchanged between coronaviruses. Recombination within the spike gene occurred more frequently than within the nonstructural region, and more commonly involved the entire domains of the spike protein [69]. Therefore, these (+)RNA viruses with very different genome organization and realization had a common general recombination pattern, which could effectively provide independent evolutionary trajectories for structural and non-structural proteins. Protein(s) function was the major factor defining their relative mobility by recombination. The authors speculate that this recombination profile may reflect the suggestive distinct evolutionary origin of these virus components billions of years ago. On the other hand, viruses of close families could have contrasting recombination patterns within these major genome blocks.

#### 2.7.3. RNAswarm: A Modular Pipeline for Differential RRI Analysis in Influenza a Virus
(by Gabriel Lencioni Lovate)

RNA proximity ligation methodologies such as PARIS, SPLASH, and 2CIMPL have offered experimental ways to detect RNA-RNA interactions (RRIs) on a large-scale [74,75,76]. However, an established bioinformatics pipeline for statistically comparing the frequency of RRIs across different strains or experimental conditions with high throughput has yet to be realized. To fill this gap, Lencioni Lovate et al. have developed RNAswarm https://github.com/gabriellovate/RNAswarm, a modular, reproducible Nextflow pipeline specifically designed for high-throughput differential RRI analysis. RNAswarm is openly accessible and is presently under active development. This tool has been successfully utilized in the analysis of SPLASH datasets of influenza A virus (IAV), a substantial global health threat due to its severe morbidity and mortality impacts. Through this application, RNAswarm has revealed differentially structured regions within IAV’s segmented RNA genome, as described in a recent publication [62].

With RNAswarm, the authors have been able to identify strain-specific RRI sites in various IAV strains, thus validating previous findings [60] and quantifying variations in prevalence of RRIs across strains [62]. The pipeline, see Figure 8, initiates by processing raw reads from RNA proximity ligation experiments, then identifies RRIs, and finally utilizes DEseq2 to sumarize differential RRI representation across distinct IAV strains or experimental conditions [77]. Additionally, a module for *de novo* annotation of discrete interactions—by generating pairwise matrices of chimeric reads and fitting Gaussian Mixture Models to identify normally distributed potential interactions is currently under development.

RNAswarm has proven successful in identifying key RRIs across replicates and in detecting highly conserved or flexible interaction sites. It offers a reliable and automated approach for prioritizing and comparing RRIs across diverse conditions or organisms. With the potential to uncover new RNA-RNA interactions in viruses and other organisms, the modularity and reproducibility of RNAswarm render it an invaluable tool for researchers investigating RNA-RNA interactions in various contexts.

### 2.8. Viral Sequence Analysis

Viral sequence analysis plays a pivotal role in understanding viral evolution, diversity, and functional characteristics. Computational tools are essential for analyzing viral sequences and extracting valuable insights. These tools encompass a range of bioinformatics algorithms and pipelines that aid in tasks such as sequence alignment, variant calling, phylogenetic reconstruction, and functional annotation. Computational approaches enable researchers to identify conserved regions, detect genetic variations, predict protein structures and functions, and explore viral-host interactions. Additionally, these tools facilitate the development of diagnostic assays, antiviral drugs, and vaccines by identifying viral targets and epitopes. Viral sequence analysis, powered by computational tools, accelerates our understanding of viral biology and informs strategies for disease surveillance, outbreak control, and therapeutic interventions. This session was hosted by Anne Kupczok (Wageningen University, The Netherlands).

#### 2.8.1. Embedding Segmented Viral Genomes for Visualisation, Search, and Clustering
(by Udo Gieraths)

Important human and animal pathogens like influenza and rotaviruses have segmented genomes. Such viruses can reassort during co-infection of a cell with different viral strains, in which case a new viral genome is created via the exchange of segments. In such cases, the evolutionary history, and therefore the phylogenetic trees of each segment, may differ considerably. This property hinders classical phylogenetic analysis and simple searches for similar genomes. Gieraths et al. presented an approach to embed segmented viral genomes in a mathematical space that allows efficient search and clustering. In the context of clustering, outliers that do not fit into any cluster are easily identified. These outliers represent rare reassortment events of particular interest, as reassortant viral genomes can give rise to highly pathogenic variants. Using the example of the influenza A virus, the authors show various applications of our developed segmented viral genome embedding in the context of search, clustering, and outlier detection.

#### 2.8.2. Hyper-EINS: A Tool for Automated Identification of Insertions in the Hepatitis E Virus Hypervariable Region (by Maximilian Nocke)

Hepatitis E virus (HEV) infections are usually asymptomatic and self-limiting, while in immunocompromised or other risk group patients may develop chronic courses [78]. The hypervariable region (HVR) within HEV’s first open reading frame is known to integrate sequence snippets of human and viral origin. Some insertions are associated with replication fitness in vitro and chronicity in vivo [79,80,81,82]. The off-label drug ribavirin is commonly used to treat chronically infected patients, but presents a high rate of treatment failures [83].

With their tool Hyper-EINS, Nocke et al. aim to provide a time efficient bioinformatics pipeline tool that automates identification and validation of insertions in high-throughput sequencing (HTS) data, while offering an easy to use graphical user interface (GUI) to simplify accessibility. Their tool will allow early identification of insertions possibly linked to treatment failure.

During Hyper-EINS development, the authors used, implemented and tested a broad variety of informatics tools and languages to increase efficiency of run time and storage usage.

A first command line only version of the analysis tool was written in Python 3 Python 3.10.12 and established the general workflow. Combined use of Cutadapt [84] and Trimmomatic [85] was applied to trim reads in an automated manner. To make read assignment sensitive to unexpected insertions, blastn was integrated, accessing NCBI nucleotide database remotely [86,87]. Identified insertions were exclusively of humane origin. Due to this, blastn was replaced with MMSeqs2 [88]. Furthermore, introduction of MMSeqs2 to Hyper-EINS reduces false positive discovery rate by evading poorly annotated data from the NCBI nucleotide database. Computation of full insertion sequences is achieved by calling the *de novo* assembler Trinity [89].

To speed up analysis processes, Hyper-EINS was reimplemented in Julia. Furthermore, exchanging blastn with MMSeqs2 reduced run time drastically and eliminated the need to access an online database, by providing a local database containing HEV and human reference genome. The tool’s Julia version features a Gtk based prototype GUI.

Hyper-EINS was validated by monitoring dynamic rearrangements in the HVR of a chronically HEV infected patient. Nocke et al. identified novel insertions of critical impact for viral fitness as proven in a subgenomic replicon system in vitro. Interestingly, content and distribution of insertions in the viral population were very dynamic in this patient, underlining an important role in HEV chronicity and treatment failure.

In conclusion, Hyper-EINS has been designed as a user-friendly tool for detecting insertions of human or viral origin in the HVR of HEV from HTS data using computers with limited RAM and processing power. The graphical user interface visualizes the pipeline’s output to make data interpretation and validation easier for the user. Early identification of HVR rearrangements in HEV infected patients can guide treatment decisions in a personalized medicine approach, based on both amplicon sequencing data specifically of the HVR or more spanned genomic regions covering the HVR. Hyper-EINS is written in Julia and will be publicly available in the future.

#### 2.8.3. Magnipore: Predicting Differential Single Nucleotide Changes in Oxford Nanopore Technologies Sequencing Signal in SARS-CoV-2 (by Jannes Spangenberg)

Oxford Nanopore Technologies (ONT) revolutionizes the field of RNA analysis by enabling direct sequencing of ribonucleic acids (RNA) and facilitating the detection of RNA modifications. RNA modifications play a crucial role in cellular processes, including gene expression regulation, RNA stability, and protein synthesis [90]. They also have a significant impact on viral infection, replication, and even the host antiviral innate immunity [91,92,93,94,95].

However, existing methods and basecallers have limitations in directly detecting and characterizing RNA modifications comprehensively. As an alternative approach, Spangenberg et al. introduced Magnipore https://github.com/JannesSP/Magnipore, see Figure 9 a novel tool designed to identify significant signal shifts in ONT data obtained from samples of closely related or similar species. By comparing the ONT signals of two samples, Magnipore aims to uncover potential differential RNA modifications, providing insights into their occurrence and distribution.

In their study, the authors applied Magnipore to analyze 16 SARS-CoV-2 samples derived from human patients and cultivated in vero cells. These samples represent a diverse range of lineages, including the early 2020s Pango lineages (n = 6), B.1.1.7 (n = 2, Alpha), B.1.617.2 (n = 1, Delta), and B.1.1.529 (n = 7, Omicron). Through the utilization of position-wise Gaussian distribution models and employing a significance threshold, Magnipore effectively identifies differential ONT signals associated with mutations between the samples and potential RNA modifications.

Notably, when investigating the Alpha and Delta variants, Magnipore detects 55 mutations and identifies 15 sites that suggest the presence of differential modifications. Across all comparisons, Magnipore achieved a mutation detection rate of 89.1%. Furthermore, Magnipore unveiled promising potential differential modifications specific to virus variants and variant groups.

In summary, Magnipore represents an advancement in the analysis of RNA modifications within the context of viruses and viral variants. By utilizing Magnipore, researchers can uncover differential RNA modifications, providing critical insights into their role in cellular processes and their potential involvement in disease development.

### 2.9. Machine Learning in Viral Surveillance

Machine learning has emerged as a valuable tool in viral surveillance, enabling the analysis of large-scale genomic and epidemiological data. Computational tools utilizing machine learning algorithms can classify and predict viral strains, identify potential outbreaks, and assess the risk of viral transmission. These tools aid in the early detection and monitoring of viral diseases, enhancing public health surveillance efforts. Machine learning also helps uncover hidden patterns and relationships in viral data, providing insights into viral evolution, host tropism, and drug resistance. Leveraging machine learning in viral surveillance empowers researchers and public health agencies to make informed decisions for effective prevention and control strategies. The session was guided by Ingrida Olendraite (Cambridge, UK).

#### 2.9.1. From High-Throughput Testing to Genomic Surveillance and Public Health Data Integration
(by Bernhard Renard)

With its open-view approach and integration with high-throughput automation, genomic sequencing plays an increasing role in infectious disease diagnostics as well as in public health surveillance programs. Facilitated by algorithmic and machine learning approaches for signal processing and information aggregation, rapid sequencing procedures are arriving in clinical settings. Focus of the work of Renard et al. has been to facilitate data analysis already during run time of second and third generation sequencers in order to speed up diagnostics and decision making as well as enrichment of target sequences [96,97].

At the same time, the authors see an increase in genomic surveillance, which allows early detection of outbreaks and prediction of spreading patterns. Thereby, it can complement more traditional epidemiological approaches [98]. Renard et al. have introduced platforms for genomic surveillance to learn and predict movement and spreading patterns in population as well as for predicting genomic risk patterns [99,100,101]. In order to further leverage learning, Graph Neural Network allow integration and prediction across heterogeneous sources.

#### 2.9.2. BLOODVIR: Virus Surveillance System for Plasma Pools Based on High-Throughput Sequencing and Machine Learning (by Martin Machyna)

Novel and re-emerging viruses pose a threat to the general public [102]. Many of these viruses are bloodborne and present an immediate risk to receivers of blood donations or blood-derived products. It is therefore vital to establish surveillance systems that are capable of detecting viruses in an unbiased manner before they can have a chance to spread in human population. Machyna et al. reported their efforts on devolvement of BLOODVIR—a virus detection system for continuous monitoring of infection risks in blood plasma using high-throughput sequencing and machine learning. BLOODVIR uses a combination of host depletion (nuclease treatment [103]) and viral enrichment (VirCapSeq-VERT [104]) to extract viral genomes and to create high-quality sequencing libraries from human blood plasma pools. Sequencing reads are classified with MiCoP [105] against a custom index created from a filtered Reference Viral Database (RVDB) [106] where genomes with very similar sequences (Mash [107] score < 0.15) were removed to reduce redundancy. The authors’ evaluation of metagenomic classification tools for virus detection identified alignment-based classifiers as more suitable than k-mer-based. While k-mer approaches tend to be faster and have higher sensitivity under normal conditions, alignment-based methods such as MiCoP performed better under conditions when investigated sequences differed significantly from reference genome due to naturally occurring mutations. In order to improve their predictions of known and novel viruses from high-throughput sequencing data, Machyna et al. performed hyperparameter tuning of DeepMicrobes [108] deep neural network (DNN) model designed for metagenomic classification. They trained the model with HIV-1 B-subtype genome fragments that were in silico generated using a range of mutation rates. Evaluation with sequences generated from other HIV variants (A, C-L, N, O, P and U) revealed a dramatic improvement in the ability to detect these ‘unseen’ HIV variants by models trained with sequences created with ≥10% mutation rate compared to model trained with non-mutated sequences. Machyna et al. tested BLOODVIR on plasma samples containing decreasing concentration of various viral standards and observed almost perfect linear relation between number of detected viral reads and virus concentration (average correlation coefficient = 0.958). In addition, their results indicate that BLOODVIR is capable of detecting as low as 100 viral genomes per mL of human blood plasma thus proving its usability for plasma pools with low viral titers. In conclusion, BLOODVIR surveillance system shows a great potential for monitoring of known and emerging viruses in human population.

#### 2.9.3. Modelling the Zoonotic Capabilities of Avian Influenza via Genomic Machine Learning
(by Liam Brierley)

Avian influenza is currently a high-risk threat in Europe. The 2021/22 outbreak was the largest yet observed and several countries stringently controlled domestic birds in response. Zoonotic capability has been observed for 14 subtypes, most recently H3N8 in 2021 [109]. Infections have so far been minimally transmissible between humans though continued zoonotic transmission events (e.g., in the UK, Spain, and Russia [110]) have increased concerns about the potential for new lineages to emerge that may spread more widely.

Several seminal modelling studies have demonstrated that machine learning algorithms can be trained directly on genome sequence data to make adequate predictions about which virus species may represent future zoonoses [101,111]. However, few models of zoonotic potential have addressed the wide variation between influenza A virus subtypes.

Brierley et al. used NCBI GenBank and GISAID to source over 18,000 whole genome sequences of avian influenza from 122 subtypes. To prevent over-fitting models to well-sampled lineages, they used MMseqs2’s Linclust algorithm [112] to collapse these to 3958 non-zoonotic clusters and 88 zoonotic clusters sharing ≥90% sequence identity across ≥80% mutual genome coverage. They selected random forest methods based on performance in previous efforts to classify influenza sequences [113].

Taking 4046 cluster-representative sequences as the training set, Brierley et al. trained random forests to distinguish zoonotic from non-zoonotic clusters based on genome composition of nucleotides, dinucleotides, and codons. Training procedures featured inner loops of 10-fold cross-validation nested within an outer loop of hold-one-out cross-validation applied to cluster representatives.

Random forest models distinguished zoonotic status with good separability (AUC = 0.95, F1 = 0.99), albeit with low absolute probabilities for some zoonotic sequences. Performance was sufficiently generalisable across subtypes, i.e., zoonotic cluster representatives predicted with the strongest confidence covered not only well-represented subtypes (e.g., H5N1, H7N9) but also poorly-represented subtypes (e.g., H10N8). The most informative model features were primarily biases in dinucleotide usage, particularly GC, AC, and CG dinucleotides.

Brierley et al. demonstrated genomic machine learning models can be tailored to identify which lineages of circulating avian influenza have potential to become zoonotic in future. These methods also have potential to highlighted weaker, unrecognised signals of viral adaptation associated with human infectivity across whole genomes. If trained appropriately, computational learning frameworks can inform more reactive strategies to prevent zoonotic infection by suggesting key genomic sites to monitor viral evolution in wildlife and generating risk estimates for newly identified viruses as soon as sequences are available [114].

### 2.10. Viral Pathogenesis

Jenna Kelly (Bern, Switzerland) hosted the session about viral pathogenesis, which refers to the mechanisms by which viruses cause disease in their hosts. Computational tools play a significant role in understanding viral pathogenesis by enabling the analysis of viral genomic data, protein structures, host-virus interactions, and immune responses. These tools help identify viral virulence factors, study the molecular basis of pathogenesis, and predict the impact of viral mutations on disease outcomes. Computational modeling and simulation assist in unraveling the complex dynamics of viral infections and aid in the development of targeted therapies and vaccines. The integration of computational tools with experimental approaches accelerates our understanding of viral pathogenesis and aids in effective disease management strategies.

#### Sex Differences in Respiratory Virus Infections
(by Sebastian Beck)

Respiratory viruses, such as influenza A viruses or coronaviruses, remain the major causative agents of acute respiratory distress syndrome (ARDS) that is associated with high morbidity and mortality. Retrospective analyses of the SARS-CoV-2 pandemic revealed old age, underlying comorbidies (e.g., obesity or diabetes) and in particular male sex as high-risk factors for severe or even fatal COVID-19 [115]. Similarly, a male bias towards severe infections was also observed for avian H7N9 influenza that emerged in China in 2013 [116]. Factors that mediate sex disparity upon respiratory virus infection may include gender aspects (e.g., social behaviour), fixed genetic predispositions, or dynamic changes in sex hormones, such as testosterone and estradiol. Beck et al. have recently shown that H7N9 avian influenza hits the metabolic HPG (hypothalamic–pituitary–gonadal) axis in men but not in women, leading to a significant reduction in circulating testosterone levels, which is associated with fatal outcome, see Figure 10 [117]. Using a mouse model for influenza infection, the authors further demonstrated a causal link between viral infection and testosterone depletion. Interestingly, men suffering from severe COVID-19 disease also presented reduced testosterone levels, as reported by us and others [118]. Strikingly, long-term monitoring of COVID-19 patients who recovered from acute infection, revealed that up to 30% still show low testosterone levels one year after recovery [119]. Collectively, these data highlight the need to further study the effect of respiratory virus infections on the endocrine system in more detail. Herein, monitoring sex hormone levels as novel biomarkers for disease severity may be crucial for individualized patient therapy in the future.

### 2.11. Metagenomics for Identifying and Tracking Potential Zoonotic Viruses

Metagenomics has emerged as a powerful tool for identifying and tracking potential zoonotic viruses, which can jump from animals to humans. By sequencing genetic material from diverse environmental samples, such as animal reservoirs or their habitats, metagenomics allows for the detection of viral sequences that may be novel or closely related to known zoonotic viruses. This approach aids in surveillance efforts, enabling the early detection of emerging viruses and facilitating proactive public health interventions to mitigate potential zoonotic disease outbreaks. Metagenomics provides a comprehensive and unbiased approach to monitor viral diversity and identify potential sources of zoonotic transmission, ultimately contributing to global health security. Jelle Matthijnssens (Leuven, Belgium) hosted this session.

#### Discovering and Tracking Potential Zoonotic Species from Metagenomic Samples with a Capture-Based Oriented Pipeline (by Maria Tarradas-Alemany)

From the dawn of Next Generation Sequencing(NGS) technologies, those strategies have become crucial in the study of microbial communities from environmental samples. However, there are still some challenges to overcome, either from biological and computational perspectives, to characterize their virome composition. Viral metagenomics has to deal with low quality sequences, possible sample biases (due to chemical inhibitors, degradation, etc.), challenging data analysis, and more specifically the lack of standardized regions for classification, the arduous purification of enough biomass for sequencing, and the limited completeness of the available virus databases. In addition, most of the viral particles found in environmental samples correspond to bacteriophages, which further complicates the detection of specific viral families and species [120].

The proposed approach to overcome some of those issues focuses on the use of capture probes specifically designed to hybridate a set of species of interest, with the aim to enrich the sample with their genomic sequences and similar ones [104]. For a specialized bioinformatic analysis of these datasets, Tarradas-Alemany et al. introduced CAPTVRED https://github.com/MarTarAl (Capture-based metagenomics Analysis Pipeline for tracking ViRal species from Environmental Datasets), a NextFlow [121] automated pipeline purposely designed to provide comprehensive results of capture-based metagenomics datasets. The pipeline includes a pre-filtering stage to discard non-viral sequences, taking advantage of a curated viral database, which also excludes phage viral sequences, as reference. Unlike other available protocols, CAPTVRED offers the flexibility to adjust almost any parameter at each step, making it adaptable to the unique characteristics of viral metagenomic datasets.

The virome present in a set of samples retrieved from sewage and bat guano have been already analyzed with this pipeline; moreover, sequences obtained by whole-genome shotgun and probe-based viral capture approaches have been also considered, in order to assess the performance of the capture kit, as well as for the pipeline. The results show an increased number of assigned viral contigs in the capture approach (using RVDB database), which also recalls higher coverage and similarity with respect to reference sequences of potentially zoonotic viruses.

The paper’s highlights include a strong emphasis on bioinformatics methods specifically designed for viurses. Most contributions of the conference address challenges in ultra-massive data analysis and an increasingly prominent role of artificial intelligence (AI) in the field. Nevertheless, the tools introduced in this publication underscore the pressing demand for further development of virus-specific bioinformatics tools, a need that has not been prominently addressed at other bioinformatics conferences. Conversely, the viruses featured in this conference have been thoroughly analyzed using specially tailored bioinformatic tools, reflecting a unique and vital synergy that does not typically find expression in virus-focused conferences.

## Figures and Tables

**Figure 1 viruses-15-02031-f001:**
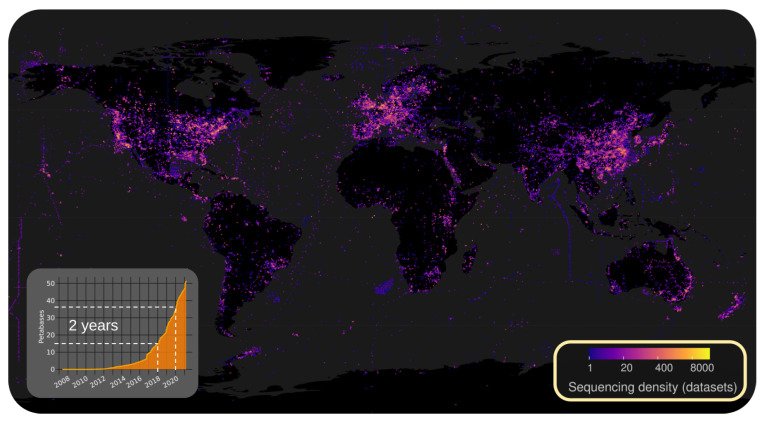
Unlocking Earth’s Virome via public data. Petabases of total (public and dbGAP) sequencing data in the Sequence Read Archive (SRA), collected over 15 years is growing exponentially (inlay) and spans the globe. Data geo-coded from SRA [16] and associated BioSamples [17] meta-data.

**Figure 2 viruses-15-02031-f002:**
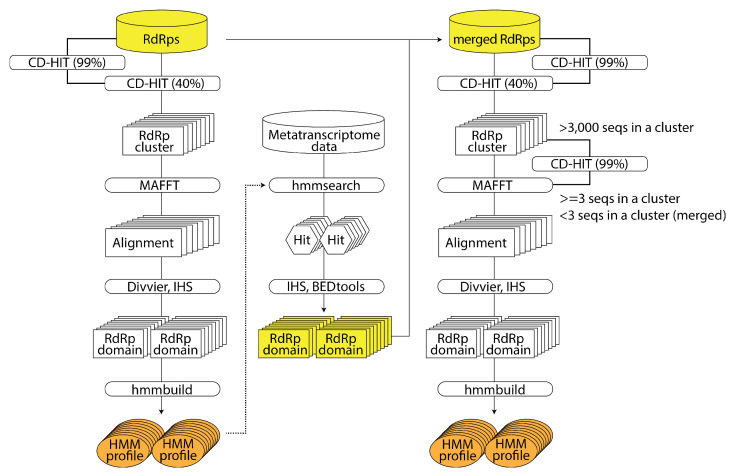
This is an overview of the pipeline utilized in NeoRdRp [22]. The input amino acid data and the resulting HMM profiles are shown in yellow and orange, respectively. First-round HMM profiles are created using curated RdRp seed datasets. These HMM profiles are then used to search for RdRp domain sequences using HMM search. Finally, second-round HMM profiles are created by the seed RdRp datasets with the obtained RdRp domain data.

**Figure 3 viruses-15-02031-f003:**
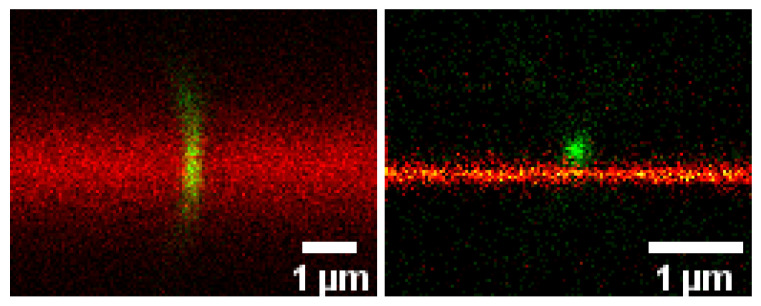
Conventional confocal (**left**) and super-resolution STED (**right**) microscopy images (side x-z view) of a SARS-CoV-2 virus like-particle (green, GFP-labelled) with a supported lipid bilayer (fire scale, DOPC lipids decorated with ACE-2, membranes labelled with fluorescent lipid analog), highlighting the improved spatial resolution to investigate molecular details in such interaction. Data taken by Ziliang Zhao (Jena).

**Figure 4 viruses-15-02031-f004:**
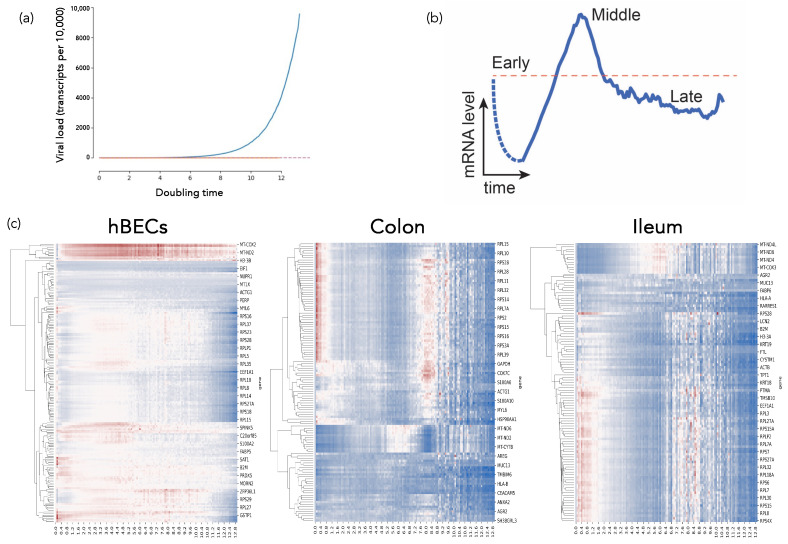
Dynamics of virus accumulation and cellular responses along infection. (**a**) Virus accumulation as a function of doubling times. Blue lines represent the number of viral transcripts in infected cells. Red lines the baseline stablished for non-infected cells. (**b**) Triphasic pattern of gene expression observed for 90% of cellular genes. (**c**) Clustering of gene expression profiles for each one of the three cell types included in the study.

**Figure 5 viruses-15-02031-f005:**
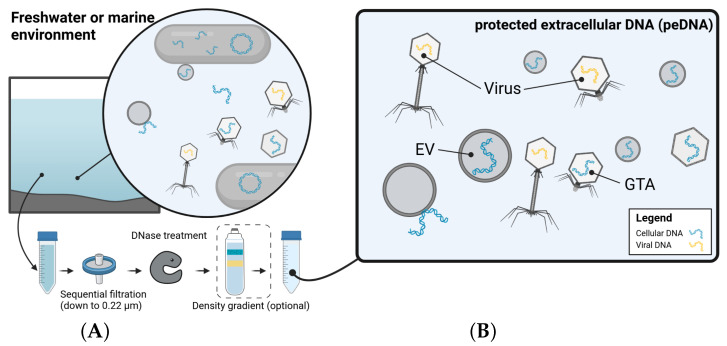
Conceptual composition of protected extracellular DNA. (**A**) Microbial entities present in a water body: microbial cells, viruses containing viral and microbial genetic material, gene transfer agents and extracellular vesicles containing host DNA. After size filtration (0.22 μm) and DNase treatment and, if applicable, purification via density gradients, microbial cells and free DNA are removed. (**B**) The remaining DNA makes up the sequence space of protected extracellular DNA, peDNA.

**Figure 6 viruses-15-02031-f006:**
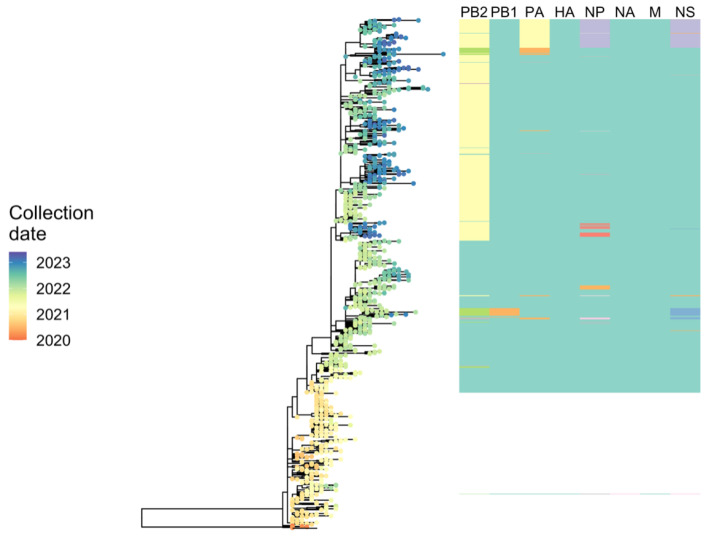
A phylogeny for HPAI H5N8 and descendant H5N1 viruses sampled in Europe since 2020 alongside schematic showing reassortment profile. Haemagglutinin phylogenetic tree generated from stratified sample of haemagglutinin nucleotide sequences sampled in Europe. Sampling date is indicated by tip node color according to the color key. To the right, a schematic shows clusters for each genomic segment for viruses of the H5N1 subtype, while the absence of such information indicates an H5N8 genome. For each segment/column, colors are assigned independently therefore the same color in two different columns should not be interpreted as indicating a shared evolutionary history between segments. Viruses with different combinations of colors across the 8 genomic segments are interpreted as arising from reassortment events.

**Figure 7 viruses-15-02031-f007:**
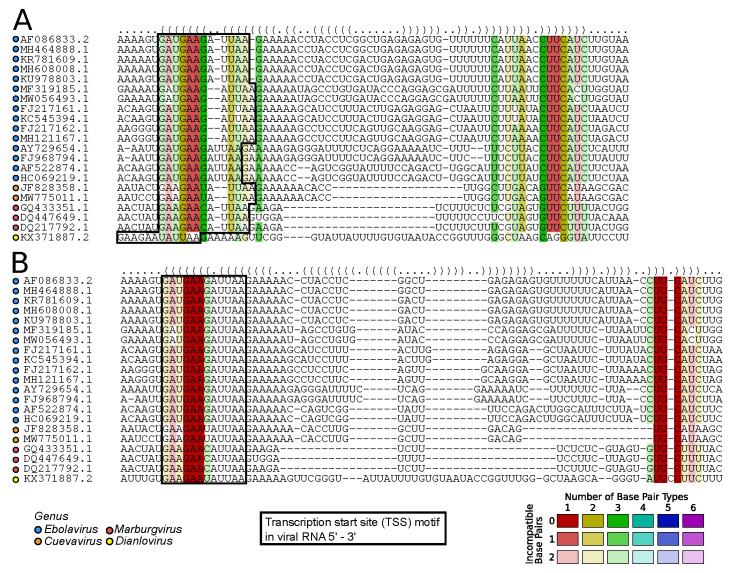
Alignment-based RNA secondary structure prediction upstream of the *VP40* gene in *Filoviridae*. (**A**) The alignment calculated using LocARNA [63] reveals structural conservation with the presence of compensatory mutations. Up to three different base pair types are observed, as indicated by the color scheme (shown in the bottom right), within the non-coding region upstream of the *VP40* gene. However, the alignment also shows disruption of the conserved transcription start site (TSS) sequence motif (illustrated by the black box). (**B**) Through manual curation, both the sequence and structure were preserved in the alignment, accurately representing the conserved elements. This refinement process led to fewer compensatory mutations, as illustrated by the base pair color scheme.

**Figure 8 viruses-15-02031-f008:**
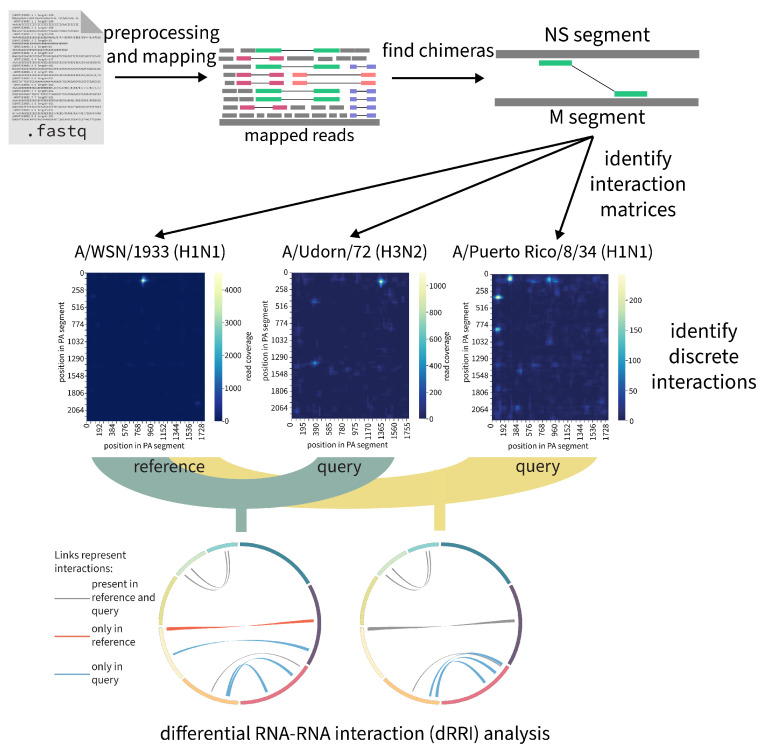
RNAswarm is a reproducible nextflow pipeline that analyzes datasets from RNA proximity ligation experiments. RNAswarm first pre-processes raw reads from RNA proximity ligation experiments, maps them to reference genomes, then identifies RRIs, and finally analyzes differential RRI representation across different strains or experimental conditions.

**Figure 9 viruses-15-02031-f009:**
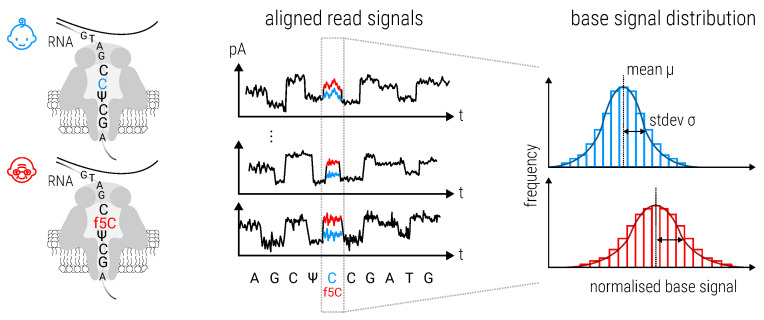
By analysing the raw signals of samples signal differences can be detected. Magnipore can read these differences and discriminate between mutations differences and possible differential RNA modifications.

**Figure 10 viruses-15-02031-f010:**
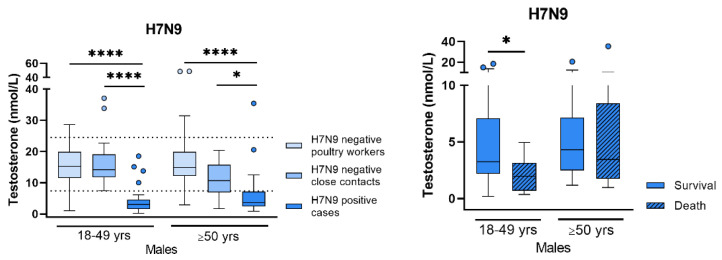
Low testosterone levels are a hallmark of severe and even fatal avian H7N9 influenza infection in men. *—*p* < 0.05; ****—*p* < 0.0001. Modified from [117].

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
