# Peer review of "The International Virus Bioinformatics Meeting 2023"

_viruses, 2023, doi:10.3390/v15102031_

Round 1
Reviewer 1 Report
The manuscript of Hufsky et al. is a conference report, and as such it fulfills the purpose. The introduction and the factual information about the conference are appropriate and provide all necessary details.
What I did not like is the way individual contributions are presented. Talk abstracts were copied to the manuscript without any change and thus keeping all usual grammatical constructs such as "we found" "we report" "we conducted" "our results" etc. I found only one case where a minor adaptation was made: while presenting the work of Terry Jones "I will present" was changed to "The talk presented".
This is not how conference reports are usually written. Instead, I would expect the authors to present an intelligent digest of the contributions and of the scientific highlights. In its present form the main bulk of the paper is simply the abstract book of the conference. I do not know whether or not this is compatible with the journal policy. If this is in line with the policy, I recommend to round up the actual text of the report with some kind of a conclusion and then to state: "Below follow the abstracts of conference talks" without any further intervention. The fact that the paper presents all the talk abstracts should also be mentioned in the paper abstract. The current statement "The report provides a summary of the key research findings and highlights shared during the meting" is not true - the report does not actually provide a summary of the key research findings and highlights.
Author Response
We thank the reviewer very much for the very valuable comments. We hope to meet now better the interests of the reviewer. With your help the manuscript is now way better readable. Thank you.
Reviewer 2 Report
NIce conference overview
Minor comments:
Later parts you use the phrase "we" often, but this is a conference overview, so change and check, should be "the authors" or something like this (happened for sure because you distributed the overview to colleagues)
Could there be a concluding discussion, placing the highlights of the meeting against alternative virus and virus bioinformatics meetings:
Which virus highlights you mentioned are particular striking and not reached in other virus conferences?
Which bioinformatics methods highlights (e.g. probably your ultra-massive data analysis or your AI parts) against other bioinformatics conferences.
Then the reader has a nice take-home message.
However, overall, very nice review and overview.
Author Response
We thank the reviewer for the valuable comments. We have included the suggestions into the manuscript and we are happy to have it better readable now. Thank you.
Round 2
Reviewer 1 Report
I am satisfied with the revision.